# THE UNREASONABLE EFFECTIVENESS OF PATCHES IN DEEP CONVOLUTIONAL KERNELS METHODS

**Louis Thiry**
Département d'Informatique de l'ENS
ENS, CNRS, PSL University
Paris, France
`louis.thiry@ens.fr`

**Michael Arbel**
Gatsby Computational Neuroscience Unit
University College London
London, United Kingdom
`michael.n.arbel@gmail.com`

**Eugene Belilovsky**
Concordia University and Mila
Montreal, Canada
`eugene.belilovsky@concordia.ca`

**Edouard Oyallon**
CNRS, LIP6, Sorbonne University
Paris, France
`edouard.oyallon@lip6.fr`

## ABSTRACT

A recent line of work showed that various forms of convolutional kernel methods can be competitive with standard supervised deep convolutional networks on datasets like CIFAR-10, obtaining accuracies in the range of $87 - 90\%$ while being more amenable to theoretical analysis. In this work, we highlight the importance of a data-dependent feature extraction step that is key to the obtain good performance in convolutional kernel methods. This step typically corresponds to a whitened dictionary of patches, and gives rise to a data-driven convolutional kernel methods. We extensively study its effect, demonstrating it is the key ingredient for high performance of these methods. Specifically, we show that one of the simplest instances of such kernel methods, based on a single layer of image patches followed by a linear classifier is already obtaining classification accuracies on CIFAR-10 in the same range as previous more sophisticated convolutional kernel methods. We scale this method to the challenging ImageNet dataset, showing such a simple approach can exceed all existing non-learned representation methods. This is a new baseline for object recognition without representation learning methods, that initiates the investigation of convolutional kernel models on ImageNet. We conduct experiments to analyze the dictionary that we used, our ablations showing they exhibit low-dimensional properties.

## 1 INTRODUCTION

Understanding the success of deep convolutional neural networks on images remains challenging because images are high-dimensional signals and deep neural networks are highly-non linear models with a substantial amount of parameters: yet, the curse of dimensionality is seemingly avoided by these models. This problem has received a plethora of interest from the machine learning community. One approach taken by several authors (Mairal, 2016; Li et al., 2019; Shankar et al., 2020; Lu et al., 2014) has been to construct simpler models with more tractable analytical properties (Jacot et al., 2018; Rahimi and Recht, 2008), that still share various elements with standard deep learning models. Those simpler models are based on kernel methods with a particular choice of kernel that provides a convolutional representation of the data. In general, these methods are able to achieve reasonable performances on the CIFAR-10 dataset. However, despite their simplicity compared to deep learning models, it remains unclear which ones of the multiple ingredients they rely on are essential. Moreover, due to their computational cost, it remains open to what extend they achieve similar performances on more complex datasets such as ImageNet. In this work, we show that an additional implicit ingredient, common to all those methods, consists in a data-dependent feature extraction step that makes the convolutional kernel *data-driven* (as opposed to purely handcrafted) and is key for obtaining good performances.

Data driven convolutional kernels compute a similarity between two images $x$ and $y$, using both their translation invariances and statistics from the training set of images $\mathcal{X}$. In particular, we focus on similarities $K$ that are obtained by first standardizing a representation $\Phi$ of the input images and then feeding it to a predefined kernel $k$:

$$K_{k,\Phi,\mathcal{X}}(x,y) = k(L\Phi x, L\Phi y),\qquad(1)$$

where a rescaling and shift is (potentially) performed by a diagonal affine operator $L = L(\Phi, \mathcal{X})$ and is mainly necessary for the optimization step Jin et al. (2009): it is typically a standardization. The kernel $K(x,y)$ is said to be *data-driven* if $\Phi$ depends on training set $\mathcal{X}$, and *data-independent* otherwise. This, for instance, is the case if a dictionary is computed from the data (Li et al., 2019; Mairal, 2016; Mairal et al., 2014) or a ZCA (Shankar et al., 2020) is incorporated in this representation. The convolutional structure of the kernel $K$ can come either from the choice of the representation $\Phi$ (convolutions with a dictionary of patches (Coates et al., 2011)) or by design of the predefined kernel $k$ (Shankar et al., 2020), or a combination of both (Li et al., 2019; Mairal, 2016). One of the goal of this paper is to clearly state that kernel methods for vision do require to be data-driven and this is explicitly responsible for their success. We thus investigate, to what extent this common step is responsible for the success of those methods, via a shallow model.

Our methodology is based on ablation experiments: we would like to measure the effect of incorporating data, while reducing other side effects related to the design of $\Phi$, such as the depth of $\Phi$ or the implicit bias of a potential optimization procedure. Consequently, we focus on 1-hidden layer neural networks of any widths, which have favorable properties, like the ability to be a universal approximator under non-restrictive conditions. The output linear layer shall be optimized for a classification task, and we consider first layers which are predefined and kept fixed, similarly to Coates et al. (2011). We will see below that simply initializing the weights of the first layer with whitened patches leads to a significant improvement of performances, compared to a random initialization, a wavelet initialization or even a learning procedure. This patch initialization is used by several works (Li et al., 2019; Mairal, 2016) and is implicitly responsible for their good performances. Other works rely on a whitening step followed by very deep kernels (Shankar et al., 2020), yet we noticed that this was not sufficient in our context. Here, we also try to understand why incorporating whitened patches is helpful for classification. Informally, this method can be thought as one of the simplest possible in the context of deep convolutional kernel methods, and we show that the depth or the non-linearities of such kernels play a minor role compared to the use of patches. In our work, we decompose and analyze each step of our feature design, on gold-standard datasets and find that a method based solely on patches and simple non-linearities is actually a strong baseline for image classification.

We investigate the effect of patch-based pre-processing for image classification through a simple baseline representation that does not involve learning (up to a linear classifier) on both CIFAR-10 and ImageNet datasets: the path from CIFAR-10 to ImageNet had never been explored until now in this context. Thus, we believe our baseline to be of high interest for understanding ImageNet's convolutional kernel methods, which almost systematically rely on a patch (or descriptor of a patch) encoding step. Indeed, this method is straightforward and involves limited ad-hoc feature engineering compared to deep learning approach: here, contrary to (Mairal, 2016; Coates et al., 2011; Recht et al., 2019; Shankar et al., 2020; Li et al., 2019) we employ modern techniques that are necessary for scalability (from thousands to million of samples) but can still be understood through the lens of kernel methods (e.g., convolutional classifier, data augmentation, ...). Our work allows to understand the relative improvement of such encoding step and we show that our method is a challenging baseline for classification on Imagenet: we outperform by a large margin the classification accuracy of former attempts to get rid of representation learning on the large-scale ImageNet dataset.

While the literature provides a detailed analysis of the behavior of a dictionary of patches for image compression (Wallace, 1992), texture synthesis (Efros and Leung, 1999) or image inpainting (Criminisi et al., 2004), we have a limited knowledge and understanding of it in the context of image classification.

The behavior of those dictionaries of patches in some classification methods is still not well understood, despite often being the very first component of many classic vision pipelines (Perronnin et al., 2010; Lowe, 2004; Oyallon et al., 2018b). Here, we proposed a refined analysis: we define a Euclidean distance between patches and we show that the decision boundary between image classes can be approximated using a rough description of the image patches neighborhood: it is implied for instance by the fame low-dimensional manifold hypothesis (Fefferman et al., 2016).

Our paper is structured as follows: first, we discuss the related works in Sec. 2. Then, Sec. 3 explains precisely how our visual representation is built. In Sec. 4, we present experimental results on the vision datasets CIFAR-10 and the large scale ImageNet. The final Sec. 4.3 is a collection of numerical experiments to understand better the dictionary of patches that we used. Our code as well as commands to reproduce our results are available here: `https://github.com/louity/patches`.

## 2 RELATED WORK

The seminal works by Coates et al. (2011) and Coates and Ng (2011) study patch-based representations for classification on CIFAR-10. They set the first baseline for a single-layer convolutional network initialized with random patches, and they show it can achieve a non-trivial performance ($\sim 80\%$) on the CIFAR-10 dataset. Recht et al. (2019) published an implementation of this technique and conducted numerous experiments with hundreds of thousands of random patches, improving the accuracy ($\sim 85\%$) on this dataset. However, both works lack two key ingredients: online optimization procedure (which allows to scale up to ImageNet) and well-designed linear classifier (as we propose a factorization of our linear classifier).

Recently, (Li et al., 2019; Shankar et al., 2020) proposed to handcraft kernels, combined with deep learning tools, in order to obtain high-performances on CIFAR-10. Those performances match standard supervised methods ($\sim 90\%$) which involve end-to-end learning of deep neural networks. Note that the line of work (Li et al., 2019; Shankar et al., 2020; Mairal, 2016) employs a well-engineered combination of patch-extracted representation and a cascade of kernels (possibly some neural tangent kernels). While their works suggest that patch extraction is crucial, the relative improvement due to basic-hyper parameters such as the number of patches or the classifier choice is unclear, as well as the limit of their approach to more challenging dataset. We address those issues.

From a kernel methods perspective, a dictionary of random patches can be viewed as the building block of a random features method (Rahimi and Recht, 2008) that makes kernel methods computationally tractable. Rudi et al. (2017) provided convergence rates and released an efficient implementation of such a method. However, previously mentioned kernel methods (Mairal, 2016; Li et al., 2019; Shankar et al., 2020) have not been tested on ImageNet to our knowledge.

Simple methods involving solely a single-layer of features have been tested on the ImageNet-2010 dataset[1], using for example SIFT, color histogram and Gabor texture encoding of the image with $K$-nearest neighbors, yet there is a substantial gap in accuracy that we attempt to fill in this work on ImageNet-2012 (or simply ImageNet). We note also that CNNs with random weights have been tested on ImageNet, yielding to low accuracies ($\sim 20\%$ top-1, (Arandjelovic et al., 2017)).

The Scattering Transform (Mallat, 2012) is also a deep non-linear operator that does not involve representation learning, which has been tested on ImageNet ($\sim 45\%$ top-5 accuracy (Zarka et al., 2019) and CIFAR-10 ($\sim 80\%$, (Oyallon and Mallat, 2015)) and is related to the HoG and SIFT transforms (Oyallon et al., 2018a). Some works also study directly patch encoders that achieve competitive accuracy on ImageNet but involve deep cascade of layers that are difficult to interpret (Oyallon et al., 2017; Zarka et al., 2019; Brendel et al., 2019). Here, we focus on shallow classifiers.

## 3 METHOD

We first introduce our preliminary notations to describe an image. A patch $p$ of size $P$ of a larger image $x$, is a restriction of that image to a squared domain of surface $P^2$. We denote by $N^2$ the size of the natural image $x$ and require that $P \leq N$. Hence, for a spatial index $i$ of the image, $p_{i,x}$ represents the patch of image $x$ located at $i$. We further introduce the collection of all overlapping patches of that image, denoted by: $\mathcal{P}_x = \{p_{i,x}, i \in \mathcal{I}\}$ where $\mathcal{I}$ is a spatial index set such that $|\mathcal{I}| = (N - P + 1)^2$. Fig. 1 corresponds to an overview of our classification pipeline that consist of 3 steps: an initial whitening step of a dictionary $\mathcal{D}$ of random patches, followed by a nearest neighbor quantization of images patches via $\mathcal{D}$ that are finally spatially averaged.

---

[1]As one can see on the Imagenet2010 leaderboard http://image-net.org/challenges/LSVRC/2010/results, and the accuracies on ImageNet2010 and ImageNet2012 are comparable.

**Whitening** We describe the single pre-processing step that we used on our image data, namely a whitening procedure on patches. Here, we view natural image patches of size $P^2$ as samples from a random vector of mean $\mu$ and covariance $\Sigma$. We then consider whitening operators which act at the level of each image patch by first subtracting its mean $\mu$ then applying the linear transformation $W = (\lambda \mathbf{I} + \Sigma)^{-1/2}$ to the centered patch. The additional whitening regularization with parameter $\lambda$ was used to avoid ill-conditioning effects.

Figure 1: Our classification pipeline described synthetically to explain how we build the representation $\Phi(x)$ of an input image $x$.

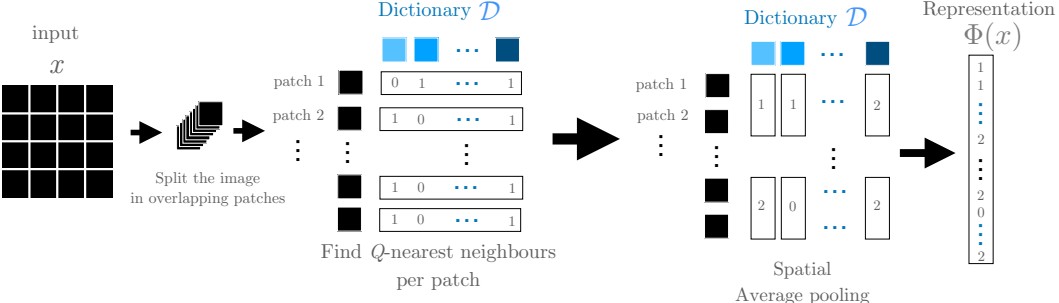

The whitening operation is defined up to an isometry, but the Euclidean distance between whitened patches (i.e., the Mahanobolis distance (Chandra et al., 1936)) is not affected by the choice of such isometry (choices leading to PCA, ZCA, ...), as discussed in Appendix A. In practice, the mean and covariance are estimated empirically from the training set to construct the whitening operators. For the sake of simplicity, we only consider whitened patches, and unless explicitly stated, we assume that each patch $p$ is already whitened, which holds in particular for the collection of patches in $\mathcal{P}_x$ of any image $x$. Once this whitening step is performed, the Euclidean distance over patches is approximatively isotropic and is used in the next section to represent our signals.

Figure 2: An example of whitened dictionary $\mathcal{D}$ with patch size $P = 6$ from ImageNet-128 (Left), ImageNet-64 (Middle), CIFAR-10 (Right). The atoms have been reordered via a topographic algorithm from Montobbio et al. (2019) and contrast adjusted.

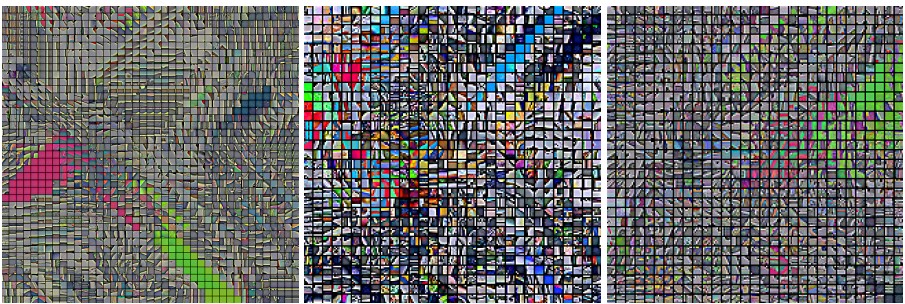

$Q$**-Nearest Neighbors on patches** The basic idea of this algorithm is to compare the distances between each patch of an image and a fixed dictionary of patches $\mathcal{D}$, with size $|\mathcal{D}|$ that is the number of patches extracted. Note that we also propose a variant where we simply use a soft-assignment operator. For a fixed dataset, this dictionary $\mathcal{D}$ is obtained by uniformly sampling patches from images over the whole training set. We augment $\mathcal{D}$ into $\cup_{d \in \mathcal{D}} \{d, -d\}$ because it allows the dictionary of patches to be contrast invariant and we observe it leads to better classification accuracies; we still refer to it as $\mathcal{D}$. An illustration is given by Fig. 2. Once the dictionary $\mathcal{D}$ is fixed, for each patch $p_{i,x}$ we consider the set $\mathcal{C}_{i,x}$ of pairwise distances $\mathcal{C}_{i,x} = \{\|p_{i,x} - d\|, d \in \mathcal{D}\}$. For each whitened patch we encode the $Q$-Nearest Neighbors of $p_{i,x}$ from the set $\mathcal{D}$, for some $Q \in \mathbb{N}$. More formally, we consider $\tau_{i,x}$ the $Q$-th smallest element of $\mathcal{C}_{i,x}$, and we define the $Q$-Nearest Neighbors binary

encoding as follow, for $(d, i) \in \mathcal{D} \times \mathcal{I}$:

$$\phi(x)_{d,i} = \begin{cases} 1, & \text{if } \|p_{i,x} - d\| \leq \tau_{i,x} \\ 0, & \text{otherwise.} \end{cases} \tag{2}$$

Eq. 2 can be viewed as a Vector Quantization (VQ) step with hard-assignment (Coates and Ng, 2011). The representation $\phi$ encodes the patch neighborhood in a subset of randomly selected patches and can be seen as a crude description of the topological geometry of the image patches. Moreover, it allows to view the distance between two images $x, y$ as a Hamming distance between the patches neighborhood encoding as:

$$\|\phi(x) - \phi(y)\|^2 = \sum_{i,d} \mathbf{1}_{\phi(x)_{d,i} \neq \phi(y)_{d,i}} . \tag{3}$$

In order to reduce the computational burden of our method, we perform an intermediary average-pooling step. Indeed, we subdivide $\mathcal{I}$ in squared overlapping regions $\mathcal{I}_j \subset \mathcal{I}$, leading to the representation $\Phi$ defined, for $d \in \mathcal{D}, j$ by:

$$\Phi(x)_{d,j} = \sum_{i \in \mathcal{I}_j} \phi(x)_{d,i} . \tag{4}$$

Hence, the resulting kernel is simply given by $K(x, y) = \langle \Phi(x), \Phi(y) \rangle$. Implementation details can be found in Appendix B. The next section describes our classification pipeline, as we feed our representation $\Phi$ to a linear classifier on challenging datasets.

## 4 EXPERIMENTS

We train shallow classifiers, i.e. linear classifier and 1-hidden layer CNN (*1-layer*) on top of our representation $\Phi$ on two major image classification datasets, CIFAR-10 and ImageNet, which consist respectively of $50k$ small and $1.2M$ large color images divided respectively into 10 and $1k$ classes. For training, we systematically used mini-batch SGD with momentum of 0.9, no weight decay and using the cross-entropy loss.

**Classifier parametrization** In each experiments, the spatial subdivisions $\mathcal{I}_j$ are implemented as an average pooling with kernel size $k_1$ and stride $s_1$. We then apply a 2D batch-normalization (Ioffe and Szegedy, 2015) in order to standardize our features on the fly before feeding them to a linear classifier. In order to reduce the memory footprint of this linear classifier (following the same line of idea of a "bottleneck" (He et al., 2016)), we factorize it into two convolutional operators. The first one with kernel size $k_2$ and stride 1 reduces the number of channels from $\mathcal{D}$ to $c_2$ and the second one with kernel size $k_3$ and stride 1 outputs a number of channel equal to the number of image classes. Then we apply a global average pooling. For the 1-hidden layer experiment, we simply add a ReLU non linearity between the first and the second convolutional layer.

### 4.1 CIFAR-10

**Implementation details** Our data augmentation consists in horizontal random flips and random crops of size $32^2$ after reflect-padding with 4 pixels. For the dictionary, we choose a patch size of $P = 6$ and tested various sizes of the dictionary $|\mathcal{D}|$ and whitening regularization $\lambda = 0.001$ . In all cases, we used $Q = 0.4|\mathcal{D}|$. The classifier is trained for 175 epoch with a learning rate decay of 0.1 at epochs 100 and 150. The initial learning rate is 0.003 for $|\mathcal{D}| = 2k$ and 0.001 for larger $|\mathcal{D}|$.

**Single layer experiments** For the linear classification experiments, we used an average pooling of size $k_1 = 5$ and stride $s_1 = 3$, $k_2 = 1$ and $c_2 = 128$ for the first convolutional operator and $k_3 = 6$ for the second one. Our results are reported and compared in Tab. 1a. First, note that contrary to experiments done by Coates et al. (2011), our methods has surprisingly good accuracy despite the hard-assignment due to VQ. Sparse coding, soft-thresholding and orthogonal matching pursuit based representations used by Coates and Ng (2011); Recht et al. (2019) can be seen as soft-assignment VQ and yield comparable classification accuracy (resp. 81.5% with $6.10^3$ patches and 85.6% with $2.10^5$ patches). However, these representations contain much more information than hard-assignment

Table 1: Classification accuracies on CIFAR-10. VQ indicates whether vector quantization with hard-assignment is applied on the first layer.

(a) One layer patch-based classification accuracies on CIFAR-10. Amongst methods relying on random patches ours is the only approach operating online (and therefore allowing for scalable training).

| Method | $|\mathcal{D}|$ | VQ | Online | $P$ | Acc. |
|---|---|---|---|---|---|
| Coates et al. (2011) | $1k$ | ✓ | × | 6 | 68.6 |
| Ba and Caruana (2014) | $4k$ | × | ✓ | - | 81.6 |
| Wavelets (Oyallon and Mallat, 2015) | - | × | × | 8 | 82.2 |
| Recht et al. (2019) | $0.2M$ | × | × | 6 | 85.6 |
| SimplePatch (Ours) | $10k$ | ✓ | ✓ | 6 | 85.6 |
| SimplePatch (Ours) | $60k$ | ✓ | ✓ | 6 | 86.7 |
| SimplePatch (Ours) | $60k$ | × | ✓ | 6 | **86.9** |

(b) Supervised accuracies on CIFAR-10 with comparable shallow supervised classifiers. Here, e2e stands for end-to-end classifier and 1-layer for a 1-layer classifier.

| Method | VQ | Depth | Classifier | Acc. |
|---|---|---|---|---|
| SimplePatch (Ours) | ✓ | 2 | 1-layer | 88.5 |
| AlexNet (Krizhevsky et al., 2012) | × | 5 | e2e | 89.1 |
| NK (Shankar et al., 2020) | × | 5 | e2e | 89.8 |
| CKN (Mairal, 2016) | × | 9 | e2e | 89.8 |

(c) Accuracies on CIFAR-10 with Handcrafted Kernels classifiers with and without data-driven reprensentations. For SimplePatch we replace patches with random gaussian noise. D-D stands for Data-Driven and D-I for Data-Independent.

| Method | VQ | Online | Depth | D-I Accuracy (D-D Improvement) | Data used |
|---|---|---|---|---|---|
| Linearized (Samarin et al., 2020) | × | ✓ | 5 | 65.6 (13.2) | e2e |
| NK (Shankar et al., 2020) | × | × | 5 | 77.7 (8.1) | ZCA |
| Simple (random) Patch (Ours) | ✓ | ✓ | 1 | 78.6 (8.1) | Patches |
| CKN (Mairal, 2016) | × | × | 2 | 81.1 (5.1)[a] | Patches |
| NTK (Li et al., 2019) | × | × | 8 | 82.2 (6.7) | Patches |

[a]This result was obtained from a private communication with the author.

VQ as they allow to reconstruct a large part of the signal. We get better accuracy with only coarse topological information on the image patches, suggesting that this information is highly relevant for classification. To obtain comparable accuracies with a linear classifier, we use a single binary encoding step compared to Mairal (2016) and we need a much smaller number of patches than Recht et al. (2019); Coates and Ng (2011). Moreover, Recht et al. (2019) is the only work in the litterature, besides us, that achieves good performance using solely a linear model with depth one. To test the VQ importance, we replace the hard-assignment VQ implemented with a binary non-linearity $\mathbf{1}_{\|p_{i,x}-d\|\leq\tau_{i,x}}$ (see Eq. 2) by a soft-assignment VQ with a sigmoid function $(1 + e^{\|p_{i,x}-d\|-\tau_{i,x}})^{-1}$. The accuracy increases by $0.2\%$, showing that the use soft-assignment in VQ which is crucial for performance in Coates and Ng (2011) does not affect much the performances of our representation.

**Importance of data-driven representations** As we see in Tab.1c, the data-driven representation is crucial for good performance of handcrafted kernel classifiers. We remind that a data-independent kernel is built without using the dataset, which is for instance the case with a neural network randomly initialized. The accuracies from Shankar et al. (2020) correspond to Myrtle5 (CNN and kernel), because the authors only report an accuracy without ZCA for this model. As a sanity check, we consider $\mathcal{D}$ whose atoms are sampled from a Gaussian white noise: this step leads to a drop of 8.1%. This is aligned with the finding of each work we compared to: performances drop if no ZCA is

Figure 3: CIFAR-10 ablation experiments, train accuracies in blue, test accuracies in red.

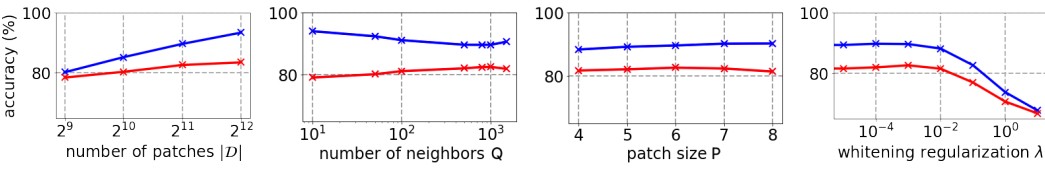

applied or if patches are not extracted. Using a dictionary of size $|\mathcal{D}| = 2048$, the same model trained end-to-end (including the learning of $\mathcal{D}$) yields to the same accuracy (- 0.1 %), showing that here, sampling patches is as efficient as optimizing them. Note that our method also outperforms linearized deep neural networks (Samarin et al., 2020), i.e. trained in a lazy regime (Chizat et al., 2019).

**Non-linear classification experiments**  To test the discriminative power of our features, we use a 1-hidden layer classifier with ReLU non-linearity and an average pooling of size $k_1 = 3$ and stride $s_1 = 2$, $k_2 = 3$, $c_2 = 2048$ and $k_3 = 7$ . Our results are reported and compared with other non-linear classification methods in Tab.1b. Using a shallow non-linear classifier, our method is competitive with end-to-end trained methods (Li et al., 2019; Shankar et al., 2020; Krizhevsky et al., 2012). This further indicates the relevance of patches neighborhood information for classification task.

**Hyper-parameter analysis**  CIFAR-10 is a relatively small dataset that allows fast benchmarking, thus we conducted several ablation experiments in order to understand the relative improvement due to each hyper-parameter of our pipeline. We thus vary the size of the dictionary $|\mathcal{D}|$, the patch size $P$, the number of nearest neighbors $Q$ and the whitening regularization $\lambda$ which are the hyper-parameters of $\Phi$. Results are shown in Fig. 3. Note that even a relatively small number of patches is competitive with much more complicated representations, such as Oyallon and Mallat (2015). While it is possible to slightly optimize the performances according to $P$ or $Q$, the fluctuations remain minor compared to other factors, which indicate that the performances of our method are relatively stable w.r.t. this set of hyper-parameters. The whitening regularization behaves similarly to a thresholding operator on the eigenvalues of $\Sigma^{1/2}$, as it penalizes larger eigenvalues. Interestingly, we note that under a certain threshold, this hyper-parameter does almost not affect the classification performances. This goes in hand with both a fast eigenvalue decay and a stability to noise, that we discuss further in Sec. 4.3.

## 4.2  IMAGENET

**Implementation details**  To reduce the computational overhead of our method on ImageNet, we followed the same approach as Chrabaszcz et al. (2017): we reduce the resolution to $64^2$, instead of the standard $224^2$ length. They observed that this does not alter much the top-performances of standard models (5% to 10% drop of accuracy on average), and we also believe it introduces a useful dimensionality reduction, as it removes high-frequency part of images that are unstable (Mallat, 1999). We set the patch size to $P = 6$ and the whitening regularization to $\lambda = 10^{-2}$. Since ImageNet is a much larger than CIFAR-10, we restricted to $|\mathcal{D}| = 2048$ patches. As for CIFAR-10, we set $Q = 0.4|\mathcal{D}|$. The parameters of the linear convolutional classifier are chosen to be: $k_1 = 10, s_1 = 6, k_2 = 1, c_2 = 256, k_3 = 7$. For the 1-hidden layer experiment, we used kernel size of $k_2 = 3$ for the first convolution. Our models are trained during 60 epochs with an initial learning rate of 0.003 decayed by a factor 10 at epochs 40 and 50. During training, similarly to Chrabaszcz et al. (2017) we use random flip and we select random crops of size 64, after a reflect-padding of size 8. At testing, we simply resize the image to 64. Note this procedure differs slightly from the usual procedure, which consists in resizing images while maintaining ratios, before a random cropping.

**Classification experiments**  Tab.2a reports the accuracy of our method, as well as the accuracy of comparable methods. Despite a smaller image resolution, our method outperforms by a large margin ( $\sim 10\%$ Top5) the Scattering Transform (Mallat, 2012), which was the previous state-of-the-art-method in the context of no-representation learning. Note that our representation uses only $2.10^3$ randomly selected patches which is a tiny fraction of the billions of ImageNet patches.

Table 2: Accuracy of our method on ImageNet.

(a) Handcrafted accuracies on ImageNet, via a linear classifier. No other weights are explicitly optimized.

| Method | $\vert\mathcal{D}\vert$ | VQ | $P$ | Depth | Resolution | Top1 | Top5 |
|---|---|---|---|---|---|---|---|
| Random (Arandjelovic et al., 2017) | - | $\times$ | - | 9 | 224 | 18.9 | - |
| Wavelets (Zarka et al., 2019) | - | $\times$ | 32 | 2 | 224 | 26.1 | 44.7 |
| SimplePatch (Ours) | $2k$ | $\checkmark$ | 6 | 1 | 64 | 33.2 | 54.3 |
| SimplePatch (Ours) | $2k$ | $\checkmark$ | 12 | 1 | 128 | 35.9 | 57.4 |
| SimplePatch (Ours) | $2k$ | $\times$ | 12 | 1 | 128 | 36.0 | **57.6** |

(b) Supervised accuracies on ImageNet, for which our model uses $\vert\mathcal{D}\vert = 2048$ patches. e2e, 1-layer respectively stand for end-to-end, 1-hidden layer classifier.

| Method | VQ | $P$ | Depth | Resolution | Classifier | Top1 | Top5 |
|---|---|---|---|---|---|---|---|
| Belilovsky et al. (2018) | $\times$ | - | 1 | 224 | e2e | - | 26 |
| Belilovsky et al. (2018) | $\times$ | - | 2 | 224 | e2e | - | 44 |
| SimplePatch (Ours) | $\checkmark$ | 6 | 2 | 64 | 1-layer | 39.4 | 62.1 |
| BagNet (Brendel et al., 2019) | $\times$ | 9 | 50 | 224 | e2e | - | 70.0 |

In Tab.2b, we compare our performances with supervised models trained end-to-end, which also use convolutions with small receptive fields. Here, $\mathcal{D} = 2k$. BagNets (Brendel et al., 2019) have shown that competitive classification accuracies can be obtained with patch-encoding that consists of 50 layers. The performance obtained by our shallow experiment with a 1-hidden layer classifier is competitive with a BagNet with similar patch-size. It suggests once again that hard-assignment VQ does not degrade much of the classification information. We also note that our approach with a linear classifier outperforms supervised shallow baselines that consists of 1 or 2 hidden-layers CNN (Belilovsky et al., 2018), which indicates that a patch based representation is a non-trivial baseline.

To measure the importance of the resolution on the performances, we run a linear classification experiment on ImageNet images with twice bigger resolution ($N = 128^2$, $Q = 12$, $k_1 = 20$, $s_1 = 12$). We observe that it improves classification performances. Note that the patches used are in a space of dimension $432 \gg 1$: this improvement is surprising since distance to nearest neighbors are known to be meaningless in high-dimension (Beyer et al., 1999). This shows a form of low-dimensionality in the natural image patches, that we study in the next Section.

## 4.3 DICTIONARY STRUCTURE

The performance obtained with the surprisingly simple classifier hints to a low dimensional structure in the classification problem that is exploited by the patch based classifier we proposed. This motivates us to further analyse the structure of the dictionary of patches to uncover a lower dimensional structure and to investigate how the whitening, which highly affects performance, relates to such lower-dimensional structure.

**Spectrum of $\mathcal{D}$** As a preliminary analysis, we propose to analyse the singular values (spectrum) of $\Sigma^{1/2}$ sorted by a decreasing order as $\lambda_1 \geq ... \geq \lambda_{d_{ext}}$ with $d_{\text{ext}} = 3P^2$ being the extrinsic dimension (number of colored pixels in each patch). From this spectrum, it is straightforward to compute the covariance dimension $d_{cov}$ of the patches defined as the smallest number of dimensions needed to explain 95% of the total variance. In other words, $d_{cov}$ is the smallest index such that $\sum_{i=1}^{d_{cov}} \lambda_i \geq 0.95 \sum_{i=1}^{d_{ext}} \lambda_i$. Fig. 4 (top) shows the spectrum for several values of $P$, normalized by $\lambda_1$ on CIFAR-10 and ImageNet-32. The first observation is that patches from ImageNet-32 dataset tend to be better conditioned than those from CIFAR-10 with a conditioning ratio of $10^2$ for ImageNet vs $10^3$ for CIFAR-10. This is probably due to the use of more diverse images than on CIFAR-10. Second, note that the spectrum tends to decay at an exponential rate (linear rate in semi-logarithmic scale). This rate decreases as the size of the patch increases (from dark brown to light brown) suggesting an increased covariance dimension for larger patches. This is further confirmed in Fig. 4(bottom-left) which shows the covariance dimension $d_{cov}$ as a function of the

Figure 4: (Top) Spectrum of $\Sigma^{1/2}$ on CIFAR-10 (top-left) and ImageNet-64 (top-right) using small patch sizes in dark-brown to larger patch sizes in light-brown. Covariance dimension (bottom-left) and nearest neighbor dimension (bottom right) as a function of the extrinsic dimension of the patches.

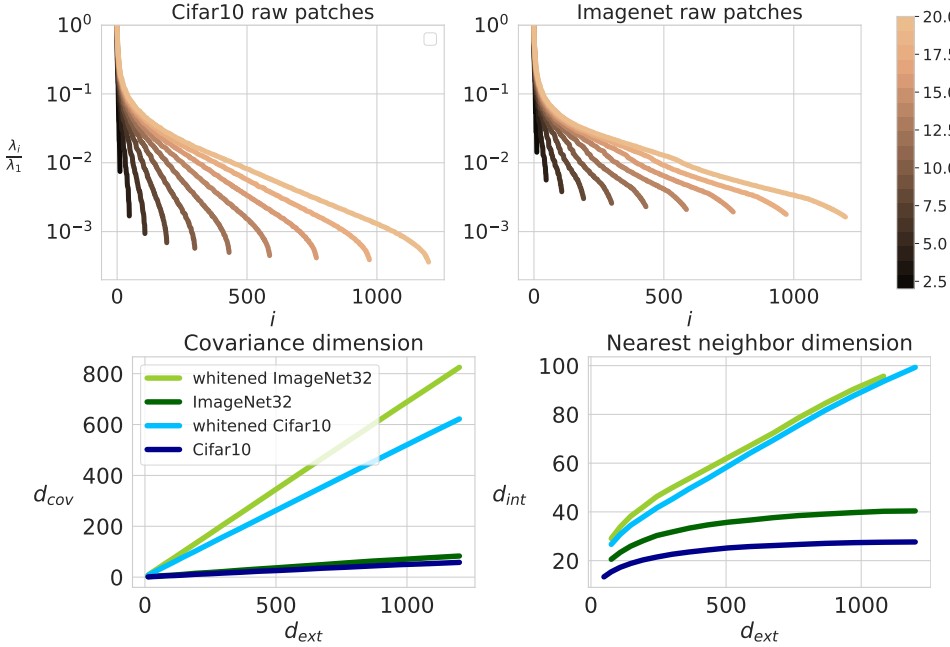

extrinsic dimension $d_{\text{ext}}$, with and without whitening. Before whitening, this linear dimension is much smaller than the ambient dimension: whitening the patches increases the linear dimensionality of the patches, which still increases at a linear growth as a function of $P^2$.

**Intrinsic dimension of $\mathcal{D}$** We propose to refine our measure of linear dimensionality to a non-linear measure of the intrinsic dimension. Under the assumption of low-dimensional manifold, if the manifold is non-linear, the linear dimensionality is only an upper bound of the true dimensionality of image patches. To get more accurate non-linear estimates, we propose to use the notion of intrinsic dimension $d_{\text{int}}$ introduced in (Levina and Bickel, 2004). It relies on a local estimate of the dimension around a patch point $p$, obtained by finding the $k$-Nearest Neighbors to this patch in the whole dataset and estimating how much the Euclidean distance $\tau_k(p)$ between the $k$-Nearest Neighbor and patch $p$ varies as $k$ increases up to $K \in \mathbb{N}$:

$$d_{\text{int}}(p) = \left( \frac{1}{K-1} \sum_{k=1}^{K-1} \log \frac{\tau_K(p)}{\tau_k(p)} \right)^{-1}. \tag{5}$$

In high dimensional spaces, it is possible to have many neighbors that are equi-distant to $p$, thus $\tau_k(p)$ would barely vary as $k$ increases. As a result the estimate $d_{\text{int}}(p)$ will have large values. Similarly, a small dimension means large variations of $\tau_k(p)$ since it is not possible to pack as many equidistant neighbors of $p$. This results in a smaller value for $d_{\text{int}}(p)$. An overall estimate of the $d_{\text{int}}$ is then obtained by averaging the local estimate $d_{\text{int}}(p)$ over all patches, i.e. $d_{\text{int}} = \frac{1}{|\mathcal{D}|} \sum_{p \in \mathcal{D}} d_{\text{int}}(p)$. Fig. 4 (bottom-right) shows the intrinsic dimension estimated using $K = 4 \cdot 10^3$ and a dictionary of size $|\mathcal{D}| = 16 \cdot 10^3$. In all cases, the estimated intrinsic dimension $d_{\text{int}}$ is much smaller than the extrinsic dimension $d_{\text{ext}} = 3P^2$. Moreover, it grows even more slowly than the linear dimension when the patch size $P$ increases. Finally, even after whitening, $d_{\text{int}}$ is only about $10\%$ of the total dimension, which is a strong evidence that the natural image patches are low dimensional.

## 5 Conclusion

In this work, we shed light on data-driven kernels: we emphasize that they are a necessary steps of any methods which perform well on challenging datasets. We study this phenomenon through

ablation experiments: we used a shallow, predefined visual representations, which is not optimized by gradient descent. Surprisingly, this method is highly competitive with others, despite using only whitened patches. Due to limited computational resources, we restricted ourselves on ImageNet to small image resolutions and relatively small number of patches. Conducting proper large scale experiments is thus one of the next research directions.

ACKNOWLEDGEMENTS

EO was supported by a GPU donation from NVIDIA. This work was granted access to the HPC resources of IDRIS under the allocation 2021-AD011011216R1 made by GENCI. This work was partly supported by ANR-19-CHIA "SCAI". EB acknowledges funding from IVADO fundamentals grant. The authors would like to thank Alberto Bietti, Bogdan Cirstea, Lénaic Chizat, Arnak Dalayan, Corentin Dancette, Stéphane Mallat, Arthur Mensch, Thomas Pumir, John Zarka for helpful comments and suggestions. Julien Mairal provided additional numerical results that were helpful to this project.

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

## A  MAHANALOBIS DISTANCE AND WHITENING

The Mahalanobis distance (Chandra et al., 1936; McLachlan, 1999) between two samples $x$ and $x'$ drawn from a random vector $X$ with covariance $\Sigma$ is defined as

$$D_M(x, x') = \sqrt{(x - x')^T \Sigma^{-1}(x - x')}$$

If the random vector $X$ has identity covariance, it is simply the usual euclidian distance :

$$D_M(x, x') = \|x - x'\| .$$

Using the diagonalization of the coraviance matrix, $\Sigma = P\Lambda P^T$, the affine whitening operators of the random vector $\mathbf{X}$ are the operators

$$w : \mathbf{X} \mapsto O\Lambda^{-1/2}P^T(\mathbf{X} - \mu), \quad \forall O \in O_n(\mathbb{R}) . \tag{6}$$

For example, the PCA whitening operator is

$$w_{\text{PCA}} : \mathbf{X} \mapsto \Lambda^{-1/2}P^T(\mathbf{X} - \mu)$$

and the ZCA whitening operator is

$$w_{\text{ZCA}} : \mathbf{X} \mapsto P\Lambda^{-1/2}P^T(\mathbf{X} - \mu) .$$

For all whitening operator $w$ we have

$$\|w(x) - w(x')\| = D_M(x, x')$$

since

$$\begin{aligned}
\|w(x) - w(x')\| &= \|O\Lambda^{-1/2}P^T(x - x')\| \\
&= \sqrt{(x - x')^T P\Lambda^{-1/2}O^T O\Lambda^{-1/2}P^T(x - x')} \\
&= \sqrt{(x - x')^T P\Lambda^{-1}P^T(x - x')} \\
&= D_M(x, x') .
\end{aligned}$$

## B  IMPLEMENTATION OF THE PATCHES K-NEAREST-NEIGHBORS ENCODING

In this section, we explicitly write the whitened patches with the whitening operator $W$. Recall that we consider the following set of euclidean pairwise distances:

$$\mathcal{C}_{i,x} = \{\|Wp_{i,x} - Wd\| \, d \in \mathcal{D}\} .$$

For each image patch we encode the $K$ nearest neighbors of $Wp_{i,x}$ in the set $Wd, d \in \mathcal{D}$, for some $K \in 1 \ldots |\mathcal{D}|$. We can use the square distance instead of the distance since it doesn't change the $K$ nearest neighbors. We have

$$\|Wp_{i,x} - Wd\|^2 = \|Wp_{i,x}\|^2 - 2\langle p_{i,x}, W^T Wd\rangle + \|Wd\|^2$$

The term $\|Wp_{i,x}\|^2$ doesn't affect the $K$ nearest neighbors, so the $K$ nearest neighbors are the $K$ smallest values of

$$\left\{ \frac{\|Wd\|^2}{2} + \langle p_{i,x}, -W^T Wd\rangle, \, d \in \mathcal{D} \right\}$$

This can be implemented in a convolution of the image using $-W^T Wd$ as filters and $\|Wd\|^2/2$ as bias term, followed by a "vectorwise" non-linearity that binary encodes the $K$ smallest values in the channel dimension. Once this is computed, we can then easily compute

$$\left\{ \frac{\|Wd\|^2}{2} + \langle p_{i,x}, W^T Wd\rangle, \, d \in \mathcal{D} \right\}$$

which is the quantity needed to compute the $K$ nearest neighbors in the set of negative patches $\overline{\mathcal{D}}$. This is a computationally efficient way of doubling the number of patches while making the representation invariant to negative transform.

## C    ABLATION STUDY ON CIFAR-10

For this ablation study on CIFAR-10, the reference experiment uses $|\mathcal{D}| = 2048$ patches, a patch size $Q = 6$ a number of neighbors $K = 0.4 \times 2048 = 820$ and a whitening regularizer $\lambda = 1e-3$, and yields $82.5\%$ accuracy. Figure 5 shows the results in high resolution. We further performed an experiment, where we replaced the patches of CIFAR-10 by the patches of ImageNet: this leads to a drop of $0.4\%$ accuracy compared to the reference model. Note that the same model without data augmentation performs about $2\%$ worse than the reference model.

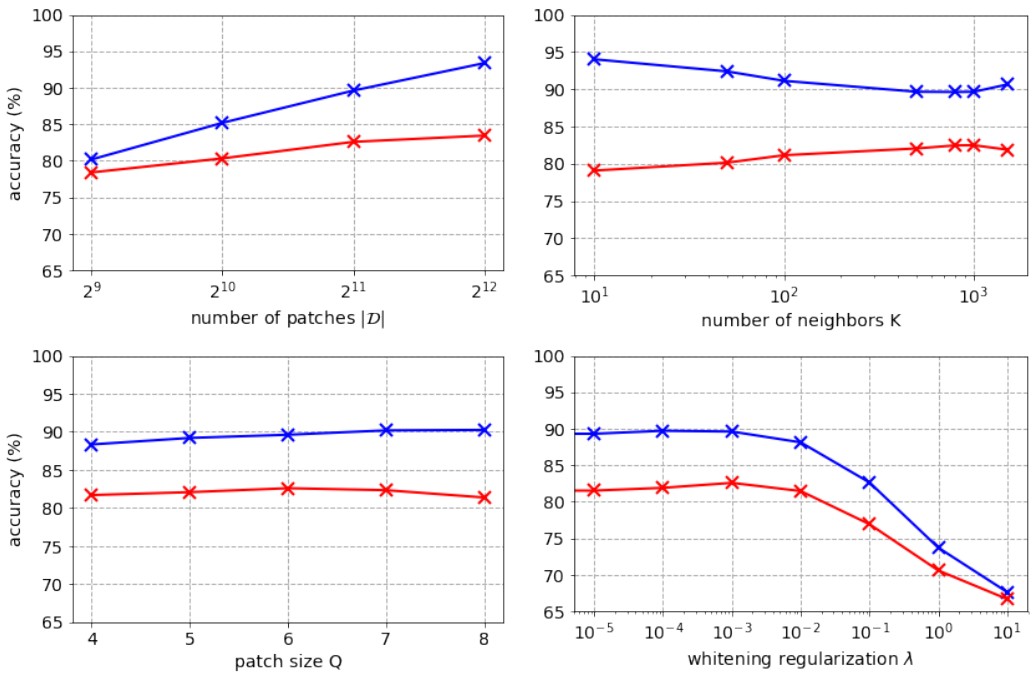

Figure 5: CIFAR-10 ablation experiments, train accuracies in blue, test accuracies in red. Number of patches $|\mathcal{D}|$ varies in $\{512, 1024, 2048, 4096\}$, number of neighbors $K$ varies in $\{10, 50, 100, 500, 800, 1000, 1500\}$, patch size $Q$ varies in $\{4, 5, 6, 7, 8\}$, whitening regularization $\lambda$ varies in $\{0, 10^{-5}, 10^{-4}, 10^{-3}, 10^{-2}, 10^{-1}, 1, 10\}$.

## D    INTRINSIC DIMENSION ESTIMATE

The following estimate of the intrinsic dimension $d_{\mathrm{int}}$ is introduced in Levina and Bickel (2004) as follows

$$d_{\mathrm{int}}(p) = \left( \frac{1}{K-1} \sum_{k=1}^{K-1} \log \frac{\tau_K(p)}{\tau_k(p)} \right)^{-1}, \tag{7}$$

where $\tau_k(p)$ is the euclidean distance between the patch $p$ and it's $k$-th nearest neighbor int the training set.

