# OpenReview forum: "The Unreasonable Effectiveness of Patches in Deep Convolutional Kernels Methods"
_ICLR.cc/2021/Conference — ICLR 2021 Poster_

### Official Review · AnonReviewer4 · 2020-10-27
**Patch based convolutional kernels - comparable results to e2e methods.**

**Rating:** 6
**Confidence:** 2

**Review:**

Summary

The authors propose using a data-driven dictionary of patches on CIFAR-10 and Imagenet, with a shallow classifier (linear / one hidden layer CNN) obtaining results comparable to end to end deeper architectures.
The proposed method consists in random sampling a dictionary of patches (2 * 10^3 for ImageNet) and using it to encode a binary representation (feature map), with Q nearest patches in the dictionary.

#####################

Rating motivation

The idea of using small patches for ImageNet and CIFAR classification is interesting - the effectiveness of small (3x3, 5x5) patches having been shown for textures in earlier work of Varma & Zisserman (A Statistical Approach to Material Classification Using Image Patch Exemplars, TPAMI 2009).
Another strong point of the paper is that it achieves an impressive accuracy (88.5) on CIFAR-10, almost on par with end-to-end AlexNet and other patch based methods, using only a small fraction of the patches obtained from ImageNet.
The clarity and readability of the paper could be improved, especially to the benefit of readers less familiar with the topic.

######################

Strong aspects of the paper:

- patch-based representation; data-driven kernels (although, it seems just random sampling)
- shallow model, with good performance;
- operates at lower resolution (64) on ImageNet, and obtains comparable performance to existing methods (trained end-to-end, and deeper).

#######################

Weaker points / questions:
- please add one introduction paragraph in Section 3, describing the method. Just introducing the notation is a bit abrupt, and makes the section difficult to read. Please try to clarify from the beginning what the one layer CNN is trained on (seems d * j binary image).

-please emphasize more the differences with Coates&Ng 2011, and compare on similar parameters (Tab 1a -- does the proposed method perform better because of using 10x , resp. 600x more patches in the dictionary, hence higher feature dimension?)

- evaluation only on CIFAR-10 and ImageNet -- could improve this by comparing with the scatter transform of Sifre & Mallat, on a common benchmark, e.g. texture / material dataset). It might be interesting, in addition to CIFAR-10 to test the method on Materials in Context (MINC) dataset, or Describable Textures Dataset (~6.5K images).

- How is the dictionary augmented? There is a mention of contrast invariance, and union over {d, -d} -- is the dictionary size |D| after augmentation?
- Please explain / clarify equation 3 - why is it necessary and what is a benefit of viewing the distance between two images as a Hamming distance? Are there computational advantages?
- Please clarify the notation in eqn. (4) - why is the step in Eq 4 necessary, and why not using a max-pooling layer in the one layer CNN?

- Please expand the explanation of the difference between data-driven (D-D) vs Non-Data-Driven (Tab 1.c); Are the patches the same for all the methods? It would also help to have the total numbers for the data-driven column (and improvement number in brackets, e.g. 78.8 ( +13.2 ).
- How would the method perform with larger patch sizes? (e.g. P=9, to have a closer comparison with BagNet).
- In Sec. 4, please add a brief explanation of spectrum definition and intrinsic dimension.

######################

Minor (typos, editing …)
- in Sec. 3(Method), first paragraph, "squared domain of surface Q^2" --> should be P^2; and patch size should be P;
- Please use 500K, 1.2M instead of 5*10^5, 1.2 * 10^6 for the number of images in CIFAR-10 and ImageNet respectively.
- Incomplete list of authors for Convolutional Kernel Methods.

---

> ### Author Response · Authors · 2020-11-13
> **Answer to AnonReviewer4**
>
> Dear Reviewer,
>
> We thank you very much for your detailed review. We address each of the points you raised below:
>
> - Introduction paragraph in Section 3: We have added a few more sentences at the beginning of Section 3, as well as the Figure 1 to clarify the model described in Section 3.
>
> - Comparison with “Coates&Ng 2011”: We believe our performances improvements were possible because we designed an online, fast training procedure that incorporates modern deep components. Thus we could use a significantly larger dictionary compared to Coates&Ng 2011 thanks to the factorization of our linear classifier. We have emphasized this point in the Related Work section: “However, both works lack two key ingredients: online optimization procedure  (which allows us to scale up to ImageNet) and well-designed linear classifier (as we propose a factorization of our linear classifier).”
>
> - Evaluation on DTD/MINC: Note that it’s very typical to test purely this kind of approach on datasets like CIFAR10 (eg, Li et al 2018, Mairal 2016, Coates 2011), and that testing it on ImageNet is a big jump. However, the points raised by the reviewer are interesting in this context because DTD or other texture datasets have a known generative process for which scattering is provably efficient. Indeed, the success of the Scattering Transforms comes from its ability to build invariances to Euclidean groups of variability, in a context for which this  geometric variability is dominant: this is clearly the case for texture discrimination, contrary to regular image classification. Incorporating this type of prior to reduce the problem of classification to a sample complexity problems would require modifications for instance of the nearest neighbour step. One could imagine a strategy where instead of considering the $d(x,y)$ metric between patches $x,y$, we would consider the quotient metric, ie $\inf_{R\in SO_2} d(x,Ry)$. We believe that adapting our method to the above setting would be an interesting future  research direction, however we also believe it to be beyond the scope of this paper.
>
> - Size $|\mathcal{D}|$: No, $|\mathcal{D}|$ corresponds to the size before augmentation (edit: as done in Recht et al, 2019 - we added a sentence to clarify)
>
> - Hamming distance: Usually, researchers employ a non quantized representation. Observing neglectable loss of performances (Cf Tab 2.a) by quantizing the representation is a surprising result that indicates that our classification performances are only sensitive to the presence of a patch or not. Furthermore, from a computational point of view, binarized vectors are faster to manipulate and easier to handle in memory.
>
> - Max Pooling: the issue with using a max pooling is that it is a non-linearity, and thus our architecture would be a neural network of depth 2. Yet, in this work, we decided to work with architecture as shallow as we can to study directly the effect of patches.
>
> - Data-driven vs non-data driven: In the previous version, we used both words: **non-data driven** and **data independent** to refer to the same method as defined in the introduction: a method that builds a representation fed to a linear classifier without using the dataset (up to a rescaling which is here only for optimization purposes).  We apologize for this confusion. We now updated the document to only use **data independent**  and we added a sentence in the text to highlight that a non data driven method  “We remind that a data-independent kernel is built without using the dataset, which is for instance the case with a neural network randomly initialized.”
> The patches aren’t the same for all the methods because we directly reported the numbers from concurrent approaches.
> We performed the modification of the Table 2 that you proposed.
>
>
> - Larger patches on Imagenet: we agree this is an interesting experiment, but that will require a substantial amount of resources. Figure 3 studies the accuracy of CIFAR10 as a function of patch size, and seems to suggest that the classification accuracies would remain close.
>
> - Sec. 4,: we have added the precise details concerning the spectrum definition and intrinsic dimension. For the spectrum, we clarified that it was simply given by the singular values of the square root of the covariance matrix $\Sigma$. For the intrinsic dimension, we now provided its expression in Equation 5  along with an intuitive explanation: a measure of how many equidistant neighbors a point can have: this depends exponentially on the dimension of the data.
>
> We have further addressed all your minor comments, thanks. Please note that the paper we cite is a followup to “Convolutional Kernel Methods” called “End-to-End Kernel Learning with Supervised Convolutional Kernel Networks” written solely by Julien Mairal. We have now also added the citation to this first paper.
>
> We thank the reviewer again for his insights and we hope we addressed all his concerns, we’d be happy to answer more questions if needed.

---

### Official Review · AnonReviewer2 · 2020-10-28
**Reviewer 2 comments**

**Rating:** 6
**Confidence:** 2

**Review:**

Briefing:

This paper mainly investigates the patch-based pre-processing for image classification. The patch-based pre-processing is done by a simple convolutional kernel constructed in a data-driven manner.
The paper achieved better or comparable performance to other more heavy comparisons using a much smaller kernel size.

Strongpoints:

The paper shows that the simple whitening procedure (by mean and covariance from training-set) of first layer weights enhances the performance, rather than using a deep kernel, which can be much efficient.


Weak points:

Analysis of the improvement: The reviewer tried to find an explainable reason for the performance improvement, although the title includes 'unreasonable'. Figure 3 seems to analyze the dictionary by covariance spectrum and intrinsic dimension, but it wasn't easy to catch the main idea. A more precise explanation of the experiments and the analysis is required.

Comments:

(1) Table 1.a shows that the proposed method achieved good performance despite the hard-assignment. What if we apply soft-thresholding to the proposed method?

(2) Experiments or discussion for the ImageNet performance with a lower number of the dictionary would be meaningful.

(3) Ablation study (data-driven or not) in ImageNet performance would strengthen the contribution of the paper (in Table 2.a?)

(4) It might be out of scope, but the reviewer could not fully catch the dictionary encoding usage just for the classification task. Is this possible to apply it to other task s.a. retrieval or other feature matching task? Then, discussion or experiments of the tasks would be required.
Or, if not, the importance of the dictionary construction on the classification task would be required.

(5) The proposed method only uses one or two layers of depth, one of the main contributions. Then, what if we use slightly more layers s.a. 3, 4, or 5? This addition of layers does not require much burden, and it would be better to add the layers if it can enhance performance.

(6) It wasn't easy to catch how the paper constructs the kernel Phi, from the method. A clearer explanation of the procedure would be appreciated.

(7) Qualitative analysis of each element of the dictionary would help readers catch the paper's contribution.

Note: The reviewer does seem to catch the main strong-points and suggestions of the paper thoroughly. The reviewer requires a clearer explanation of the contribution with the replies for the above comment, and it would be required for the precise rating.

---

> ### Author Response · Authors · 2020-11-13
> **Answer to AnonReviewer2**
>
> Dear Reviewer,
>
> We thank you very much for your review. We address each of the points you raised below:
>
> Analysis of the improvement:
> Thank you for pointing out that the message from Figure 3 was unclear. We have added a paragraph at the beginning of Section 4.3 to clarify the interpretation of the results:
> “The performance obtained with the surprisingly simple classifier hints to a low dimensional structure in the classification problem that is exploited by the patch based classifier we proposed. This motivates us to further analyse the structure of the dictionary of patches to uncover a lower dimensional structure and to investigate how  the whitening, which highly affects performance, relates to such lower-dimensional structure.”
>
> In Figure 3, we estimated the spectrum (Singular values) of the square root of the covariance matrix of patches of images and provided estimates of the intrinsic dimension of those patches which turns out to be much smaller than the ambient/extrinsic dimension  (number of pixels in the patch). This suggests that image patches are indeed low dimensional. We then observed that whitening the patches has an effect on this intrinsic dimension: Whitened patches tend to have higher intrinsic dimension than non-whitened ones, both of which are much smaller than the   extrinsic dimension. This suggests that whitening helps increasing/inflating data along some dimensions that are important for classification. To clarify this we have added more descriptions of the experiments of Figure 3 in section 4.3.
> In conclusion: this analysis supports one of our key messages which is that the task of classifying images is lower dimensional than one would think, which is a surprising result.
>
> Contribution:
> A major contribution of this work is to observe that for designing competitive kernels for image classification, it is absolutely necessary to make the kernel dependent on  data. For instance, kernels carefully designed with wavelets lead to significantly worse performances than our simple approach. Another contribution of this work is to extend the work of (Saxe et al, 2011) and to show that even simple patch based procedures are competitive on ImageNet, with few learning. This indicates low dimensional structures that we further study in this work.
>
>
> 1. In Tab 2a, we used on ImageNet a soft-assignment, and we show that it only leads to a minor performance improvement.
> 2. Do you mean, with less patches? We believe the accuracy will follow the same trend as observed on CIFAR. Another interesting  question would be to actually use more patches but unfortunately we do not have the appropriate machinery.
> 3. We agree with the reviewer yet ImageNet experiments require a significant amount of resources. Thus, we decided to only conduct ablation experiments on CIFAR as many standard works have done: in general, ablation studies are done on a smaller dataset(to avoid overfitting) before running it on a larger scale dataset.
> 4. It is absolutely possible to use this representation for other tasks, as at the end, for a given image x, one has a vector $\Phi x$ that could be used somewhere else. We however decided to focus this work on supervised classification. Probably some hyper parameters will have to be tuned, but, as you pointed out, we believe this is beyond the scope of this paper.
> 5. One of the purposes of this work is to keep the architecture shallow, and to see how far one can solve the image classification task with significant constraints (no depth, no learning). Thus, we respectfully believe this experiment is beyond the scope of this paper.
> 6. Thank you for pointing this out, we have added Figure 1 to describe our architecture better.
> 7. We have provided in Figure 2 a visualization of the atoms of the dictionary which illustrates the similarity between patches and highlights their low dimensionality. We are also happy to consider alternative visualizations if suggested by the reviewer.
>
> We thank the reviewer again for his insights and we hope we addressed all his concerns, we’d be happy to answer more questions if needed.

---

### Official Review · AnonReviewer1 · 2020-10-28
**Interesting results, but still have questions for motivation**

**Rating:** 6
**Confidence:** 3

**Review:**

This paper proposes a powerful non-learning Kernal based baseline for ImageNet classification. The proposed non-learning Kernal based baseline (which can be interpretable to a vector quantization) shows comparable results (88.5) with AlexNet (89.1) in CIFAR-10 top-1 accuracy. The ImageNet result (39.4) shows that it is still challenging to classify the images without deep features, but about 40% is an impressive baseline without any learning method (e.g., these results is almost comparable to BagNet top-5 error).

**Pros**

This paper shows that only using a shallow kernel can often be comparable to deep models, e.g., AlexNet.

**Cons and questions**

**[Motivation to the shallow method]**
I wonder about the motivation of the proposed kernel-based non-learning method. Why we need a shallow kernel method while a deep data-driven method shows a significantly better result?

**[Runtime comparison]**
I would believe that if the authors can provide the number of learnable parameters, the number of flops, and the real latency time compared to the deep models can bring a huge practicalness to the real-world applications.

**[More analysis would be expected]**
Apart from the accuracies, I would expect more analysis on the patch-side. For example, BagNet paper [1] shows that ImageNet classifiers are actually heavily relying on the local cues, e.g., fingers for Tench classes.
- [1] W. Brendel, M. Bethge, and . . Approximating cnns with bag-of-local-features models works surprisingly well on imagenet. ICLR 2019

It will be interesting to many readers if the authors can provide additional analysis on the "important" patches for the image classification tasks. Explainability analysis, as Brendel et al. could be useful for many researchers.

**[Additional questions]**
It may not be related to the original task, but I wonder a few things about the proposed method.

- Is the proposed method robust to the adversarial perturbation? I notice that it could be difficult to attack the proposed method (because there is no gradient). It is okay to test with a black-box attack, e.g., generate an attacked image with well-known architectures, e.g., ResNet, and test the attacked accuracy.
- Similarly, I wonder the proposed method is beneficial to the out-of-distributed (OOD) robustness, and other distribution shifts
  - Hendrycks, Dan, and Kevin Gimpel. "A baseline for detecting misclassified and out-of-distribution examples in neural networks." ICLR 2017.
  - Hendrycks, Dan, and Thomas Dietterich. "Benchmarking neural network robustness to common corruptions and perturbations.", ICLR 2019.
  - Hendrycks, Dan, et al. "Natural adversarial examples." arXiv preprint arXiv:1907.07174 (2019).


Please consider to add "explainability analysis", "adversarial robustness", "distribution shift robustness" and "OOD robustness" to show the effectiveness of the paper.

---

Post-rebuttal review

In my initial review, my main concern was the motivation for the patch-based classification is unclear, and I asked some questions related to the further potential usage of the proposed method, particularly focusing on robustness.

During the rebuttal process, the authors address most of my concerns well in their responses.

- Main motivation: this work more focuses on the mathematical analysis of the image classification tasks, rather than performances (accuracy, runtime, ...). It makes sense to me, and I think this motivation needs to be encouraged to explore by many researchers.
- Runtime or learnable parameters: tracking the runtime comparing to deep methods is non-trivial as the author clarified. However, the authors showed that the number of learnable parameters is much less than deep models (580k for the proposed method, a few M for AlexNet). I think this comparison fairly shows that the proposed method is efficient than deep models in terms of the number of parameters.
- Other analysis that can support the motivation of this paper: I'd expect to see more robustness analyses such as the black box adversarial attack results, but I agree that this is out-of-scope of this work. To me, the responses to the additional comments are not helpful, but I understand that my questions can be out-of-scope of this work.

Hence, I'd like to change my score from 5 to 6.

---

> ### Author Response · Authors · 2020-11-13
> **Answer to AnonReviewer1**
>
> Dear Reviewer,
>
> We thank you very much for your detailed review. We hope the following answers your questions:
>
> [Motivation to the shallow method] This is an important question that you pointed out and that also motivated several prior works like (Shankar et al 2019) or (Li et al 2019). The main motivation here is to isolate the possible reasons that are behind the many successes in image classification so that a mathematical understanding is more accessible. As noted in (Mallat, Understanding deep convolutional neural networks, 2016) both the depth and the gradient descent make the mathematical analysis difficult, in particular because the problem is highly non convex. On the other hand, kernels methods are powerful tools to obtain mathematical guarantees in a classification context.
> Our work shows that a simple shallow kernel method, when using a kernel that is convolutional and data-driven achieves competitive performance. Given that kernel methods are far more amenable to a mathematical analysis, we hope this opens up future research to provide a mathematical understanding of the properties in the data that allows those simple shallow methods to work so well.
>
>
> [Runtime] The number of learnable parameters of the method of the ablation study on CIFAR-10 with 2048 patches is around 580k (compared to e.g. ~few M with AlexNet), which is mainly thanks to a cheap fully connected layer. The exact latency time is a bit harder to track because we highly exploited the parallelization possibilities: indeed, each convolution with a patch can be processed independently since there is no back-propagation. The computational complexity of our first layer is about the standard convolution complexity, ie O(P^2N^2*|D|), where |D| is the size of the dictionary, N the size of an image and P the patch size.
>
> [More analysis]:
> - Critical patches. In our work we have attempted to study which patches are important for classification purposes. However we found a sparse set of patches is not able to solve the task. Indeed it appears the ability to describe the proximity of a single patch to a potentially larger number of training set patches allows for increased classification.
> - Links to BagNet. Indeed, the work of Brendel et al can suggest that global information is less important for the final classification task than previously thought. The explainability analysis in their case further allows to analyse the complex non-linear (yet local) CNN used for classification.  Our approach uses a simpler representation for which other types of analysis are possible such as the ones we performed in section 4.3 and the ablation studies.
> - Is the proposed method robust to the adversarial perturbation? Regarding robustness, the positive results of https://arxiv.org/abs/2009.14444, that are aligned to our model, indicate that over parametrization of shallow networks is necessary to obtain robustness. Given that one needs a significant amount of patches to obtain good performances, we believe we are in this setting. Furthermore, there may be opportunities to further bound this using a model of the natural patch distribution compared to learned filters. However, this is outside of the scope of the current work.
> - out-of-distribution. This is an interesting point that we decided to explore and we included the results in the Appendix. Indeed, although the patches are data dependent, the visualizations in Fig 2 suggests that these typically extract generic features, furthermore the model is shallow. All this suggests it may be less biased than deep networks to spurious correlations in the data. To study this, we performed a simple experiment where we used the patches of Imagenet and trained a model on CIFAR10. We report an accuracy of 81.9 % (drop of 0.4%), which shows this method is robust to this.
>
> We thank the reviewer again for his insights and we hope we addressed all his concerns, we’d be happy to answer more questions if needed.

---

### Official Review · AnonReviewer3 · 2020-11-01
**A few inaccuracies but nice to see improvement on finite dimensional kernel approximations on Cifar-10**

**Rating:** 7
**Confidence:** 5

**Review:**

My understanding is that this paper extends the approach from https://www-cs.stanford.edu/~acoates/papers/coatesng_nntot2012.pdf to use *patches* from an image as filters

I enjoyed this paper quite a bit and the numbers are quite impressive (88.5 for a 2 layer network on Cifar-10 is great), and its good to understand these methods fall short on ImageNet, this may make sense for a 2 layer as the spatial pooling destroys a lot of structure. It is interesting to see these methods outperform a scattering transform. I enjoyed the rigorous ablations and study of the spectrum of patches. These are very good baseline experiments that make a valuable data point for the community. I also think that doing an ImageNet experiment at the full 224 resolution may be worth it, if that reaches 60% top-1 or above it would be a very big contribution.

Few Questions/Clarifications:

1. You seem to say that the kernel from Shankar et al is an "End to End" classifier, but from my understanding from reading that paper and looking at the corresponding code is that they just train a linear classifier on the kernel matrix which is a deterministic function, it has several layers (within in the kernel function), but the only thing thats learned is the weights that multiply the kernel matrix (the last layer), so I think in your characterization is somewhat incorrect, but please do correct me if I am wrong in my understanding of Shankar et al.


2. How many patches were used to achieve 88.5? It seems this number is missing in Table 2b.

3. Do you know what accuracy you get when you only do Flip augmentation as opposed to crops + flips, the shankar et al number you compare to only does Flips as I see. CKN seems to do no augmentation.

4. In the related work, what do you mean by "well-designed linear classifier"?

5. How does SimplePatch perform without nearest neighbors or visual quantization?

---

> ### Author Response · Authors · 2020-11-13
> **Answer to AnonReviewer3**
>
> Dear Reviewer,
>
> We thank you for your review and comments. We address your questions in order:
> 1. You are absolutely correct in your description of Shankar et al. They further incorporate in the Sec 4 of their paper a “Myrtle 5 CNN” performance that we report and which corresponds to the analogue of their kernel methods. We clarified in the revised version  that we compared to this model  by saying: “The accuracies from \cite{shankar2020neural} correspond to Myrtle5 (CNN and kernel), because the authors only report an accuracy without ZCA for this model.”.
> 2. We have added this number to the Table for clarity, in the text we reported that we used 2048 patches.
> 3. Using the setting of our ablation study, the accuracy we get without data augmentation is 80.3% (drop of 2%). Indeed, the accuracies are more difficult to compare due to the lack of data augmentation in several works. This is one of the reasons why we report the relative accuracy improvement in Table 3 of data driven over non data driven kernel, as we believe that this quantity is more stable to data augmentation. Note also that incorporating data augmentation in this type of pipeline is a contribution of this work, as it remains challenging to use data augmentation with kernel methods because of scalability. We have included this accuracy in the Appendix.
> 4. Here, a linear classifier would correspond to a large fully connected layer. Yet, we factorize this classifier in two smaller linear classifiers and incorporate some average pooling. While this still yields a linear classifier (no linearities are used in the factorization),  it  is important both for scalability and performances, we have added a reference in the text: “ well-designed linear classifier (as we propose a factorization of our linear classifier).”
> 5. In Table 2a, we report a variant that uses a soft-assignment, described in the 2nd paragraph of Sec 4.1: it gets around 57.6 top5, which is only a minor improvement compared to the quantized version.
>
> We also agree with the reviewer that it would be extremely interesting to perform the experiments on ImageNet with a large number of patches and full resolution images, but unfortunately we leave this question open due to a lack of resources: we believe it is a promising direction for future works.
>
> We’d be happy to answer more questions if needed.

---

### Author Response · Authors · 2020-11-16
**Revised version of the submission**

We have updated a third revised version, to include additional clarifications:
In the second paragraph of section 4., we clarified the notations ($\lambda_i$ and $d_{cov}$) used in the legend of figure 4.
$\lambda_i$ refers to the singular values of $\Sigma^{\frac{1}{2}}$ sorted by decreasing order and $d_{cov}$ refers to the covariance dimension at $95%$, defined as the smallest index such that $\sum_{i=1}^{d_{cov}}\lambda_i \geq .95 \sum_{i=1}^{d_{ext}}\lambda_i $.   We also fixed a typo in the labels of figure 3:  "number of neighbors $Q$" instead of "number of neighbors $K$".
We apologizes for any confusion this might have created.
We thank you for you time.

---

### Decision · Program_Chairs · 2021-01-07
**Final Decision**

**Decision:**

Accept (Poster)

**Comment:**

This paper studies the patch-based convolutional kernels for image classification, and finds that making the kernel dependent on data is necessary for designing competitive kernels for image classification. The proposed simple method shows comparable results to those end-to-end deeper architectures on CIFAR-10 and ImageNet datasets.

All reviewers feel that the paper is interesting, important, and the performance is impressive. During the rebuttal, the authors have addressed most of the questions and concerns raised by the reviewers. In particular, authors have clarified the motivation, discussed the model size of the proposed method (requested by R1), added precise details about the spectrum definition and intrinsic dimension (requested by R4), and taken the suggestions from all reviewers to improve their paper.

After rebuttal, all reviewers agree on accepting the paper. After checking the discussions between the authors and reviewers, I am convinced that the original concerns of the reviewers are addressed. Hence, I recommend that this paper be accepted.